# Echocardiographic Assessment of Right Ventricular–Arterial Coupling in Predicting Prognosis of Pulmonary Arterial Hypertension Patients

**DOI:** 10.3390/jcm10132995

**Published:** 2021-07-05

**Authors:** Remigiusz Kazimierczyk, Ewelina Kazimierczyk, Malgorzata Knapp, Bozena Sobkowicz, Lukasz A. Malek, Piotr Blaszczak, Katarzyna Ptaszynska-Kopczynska, Ryszard Grzywna, Karol A. Kaminski

**Affiliations:** 1Department of Cardiology, Medical University of Bialystok, 15-089 Bialystok, Poland; remigiuszk1989@gmail.com (R.K.); eidzkowska@wp.pl (E.K.); malgo33@interia.pl (M.K.); sobkowic@wp.pl (B.S.); kasia.ptaszynska@op.pl (K.P.-K.); 2Department of Epidemiology, Cardiovascular Disease Prevention and Health Promotion, National Institute of Cardiology, 04-628 Warsaw, Poland; lukasz.a.malek@gmail.com; 3Department of Cardiology, Cardinal Wyszynski’ Hospital, 20-718 Lublin, Poland; blaszcz12345@interia.pl (P.B.); grzywna@gmail.com (R.G.); 4Department of Population Medicine and Lifestyle Diseases Prevention, Medical University of Bialystok, 15-089 Bialystok, Poland

**Keywords:** echocardiography, coupling, pulmonary arterial hypertension, prognosis, right ventricle

## Abstract

In response to an increased afterload in pulmonary arterial hypertension (PAH), the right ventricle (RV) adapts by remodeling and increasing contractility. The idea of coupling refers to maintaining a relatively constant relationship between ventricular contractility and afterload. Twenty-eight stable PAH patients (mean age 49.5 ± 15.5 years) were enrolled into the study. The follow-up time of this study was 58 months, and the combined endpoint (CEP) was defined as death or clinical deterioration. We used echo TAPSE as a surrogate of RV contractility and estimated systolic pulmonary artery pressure (sPAP) reflecting RV afterload. Ventricular–arterial coupling was evaluated by the ratio between these two parameters (TAPSE/sPAP). In the PAH group, the mean pulmonary artery pressure (mPAP) was 47.29 ± 15.3 mmHg. The mean echo-estimated TAPSE/sPAP was 0.34 ± 0.19 mm/mmHg and was comparable in value and prognostic usefulness to the parameter derived from magnetic resonance and catheterization (ROC analysis). Patients who had CEP (*n* = 21) had a significantly higher mPAP (53.11 ± 17.11 mmHg vs. 34.86 ± 8.49 mmHg, *p* = 0.03) and lower TAPSE/sPAP (0.30 ± 0.21 vs. 0.43 ± 0.23, *p* = 0.04). Patients with a TAPSE/sPAP lower than 0.25 mm/mmHg had worse prognosis, with log-rank test *p* = 0.001. the echocardiographic estimation of TAPSE/sPAP offers an easy, reliable, non-invasive prognostic parameter for the comprehensive assessment of hemodynamic adaptation in PAH patients.

## 1. Introduction

Pulmonary arterial hypertension (PAH) is a rare disease that causes increased pulmonary vascular resistance (PVR) and elevated pulmonary arterial pressures (PAPs). This often leads to heart failure and premature death [1]. Due to the progressive course of the disease, maladaptive right ventricular (RV) dilatation and/or hypertrophy occurs, eventually resulting in RV failure [2,3]. The development of RV failure carries a two- to threefold increase in risk of cardiac death, irrespective of the degree of left ventricular (LV) dysfunction [4,5].

When the RV becomes overloaded in PAH, it mostly adapts to the increased afterload by increasing contractility according to Frank–Starling law. However, with the depletion of systolic function, the RV becomes uncoupled from pulmonary circulation and dilates to preserve flow output [2]. The best method (gold standard) to assess cardiac contractility is pressure–volume loop-derived end-systolic elastance (Ees). RV–arterial coupling is then calculated as the ratio of end-systolic elastance to arterial elastance (Ees/Ea) [2,3]. However, obtaining Ees and Ea with the use of pressure–volume loops is time consuming and requires an invasive procedure. There is still a need for accurate, noninvasive measurements of RV function, which might provide prognostic information useful for finding the group of patients needing therapy escalation, even before clinical deterioration. Previous studies confirm several noninvasive parameters of RV function as important prognostic factors in PAH, e.g., echocardiographic tricuspid annular plane systolic excursion (TAPSE); pulmonary arterial systolic pressure (sPAP) [4,5,6]; MRI-derived right ventricle ejection fraction; and standardized glucose uptake of RV in PET analysis [7]. The ratio of TAPSE/sPAP, introduced earlier, has been proposed as an index of in vivo RV shortening in the longitudinal axis versus developed force in patients with chronic heart failure, and was found to be a non-invasive, indirect measurement of RV contractile function and RV–arterial coupling [6,8,9]. It seems that this approach to the estimation of RV–arterial coupling may have functional and prognostic relevance, especially at bedside.

In this study, we aimed to (1) verify the relationship between the echo-derived RV–arterial coupling surrogate, TAPSE/sPAP ratio, and hemodynamic parameters obtained from right heart catheterization (RHC) and magnetic resonance imaging (MRI) in PAH patients, and (2) investigate whether the echocardiographic TAPSE/sPAP ratio has prognostic significance in PAH patients.

This paper presents an echo-focused analysis (with 58 months of follow-up observation) of a PAH group enrolled to the project, with a grant entitled “The role of PET/MRI hybrid imaging in assessment of pulmonary arterial hypertension patients”, of which partial results were previously published [7,10,11].

## 2. Materials and Methods

### 2.1. Population Characteristics

Into the study we enrolled twenty-eight clinically stable PAH patients. Pre-capillary pulmonary hypertension was confirmed by RHC (mean pulmonary artery pressure (mPAP) ≥25 mmHg, pulmonary artery wedge pressure (PAWP) ≤15 mmHg). According to European guidelines [1], we ruled out secondary PH causes. The exclusion criteria were the following: age below 18 years old, IV WHO class, PAH associated with prevalent systemic-to-pulmonary shunts due to moderate to large defects (according to European guidelines) [1], group II, III, IV, V of pulmonary hypertension, Eisenmenger physiology and contraindications to cardiac MRI. The control group consisted of twelve healthy controls who were matched based on sex and age. During the baseline evaluation, we performed a physical examination, six-minute walk test, cardiopulmonary exercise test (study group) and laboratory tests, e.g., serum B-type natriuretic peptide (BNP).

The clinical follow-up lasted 58 months. Death, WHO class worsening, and hospitalization due to pulmonary hypertension or right heart failure (requiring therapy with intravenous diuretics) were used as a combined clinical endpoint (CEP) for Kaplan–Meier analysis (as described before [7,10,11]).

The study was approved by the local Bioethics Committee. All patients gave written, informed consent for participating in the study, including the taking and storage of blood samples. The study complied with the Declaration of Helsinki.

### 2.2. Echocardiography

Echocardiographic examinations were performed in both study and control groups. Tricuspid annular plane systolic excursion (TAPSE) was obtained by M-mode imaging in the apical four-chamber view centered on the right ventricle. It was defined as the difference between the displacement of the RV lateral annulus from end-diastole to end-systole. The echo estimation of the sPAP (RVSP/esPAP) (reflecting RV afterload) was based on the sum of the peak velocity of tricuspid regurgitation (Bernoulli equation, TRPG = 4[TRmax]^2^) and of the estimated central venous pressure obtained by inferior cava vein diameter and collapsibility [12]. The echocardiographic mPAP was approximated from the esPAP using Chemla’s formula: emPAP = 0.61 × esPAP + 2 mmHg, as this formula had a good correlation and acceptable limits of agreement, as shown in [13,14,15].

### 2.3. RHC/MRI Hemodynamic Parameters

Right-sided heart catheterization (RHC) was performed using a Swan–Ganz catheter with standard technique [7], and during the procedure, several hemodynamic parameters were measured or calculated: pulmonary artery wedge pressure (PAWP); systolic, diastolic, and mean pulmonary artery pressure; cardiac output (calculated by thermodilution method); cardiac index; SV (obtained by dividing cardiac output by heart rate); right atrial pressure; and PVR, calculated as: (mean pulmonary artery pressure—PAWP)/cardiac output.

Cardiac magnetic resonance imaging is the gold standard noninvasive method for RV functional assessment. We used MRI scans to obtain TAPSE as a reference method to echo TAPSE [5]. MRI imaging was performed with a 3T Biograph mMR hybrid system (Siemens, Healthcare Erlangen, Erlangen, Germany). Systolic function assessment was based on SSFP short axis images from the tricuspid valve insertion point to the apex to encompass the entire RV. MRI and RHC were performed within a median of 4 days (2–6 days) in the PAH group at baseline visit.

### 2.4. Statistical Analysis

The data are expressed as a mean ± standard deviation (SD) or median (interquartile range) in the case of variables with non-normal distribution (Kolmogorov–Smirnov test). The Student’s *t*-test was used for continuous data and a χ^2^ test was used for categorical variables. Spearman’s correlation coefficient was used to examine the relationship between two continuous variables. Receiver operator characteristic curves (ROCs) were plotted to determine the area under the curve (AUC) and the sensitivity and specificity of the optimal cut-offs. To investigate the occurrence of clinical endpoints, the Kaplan–Meier method with the log-rank test was implemented; *p* < 0.05 was deemed statistically significant. A statistical software package, STATA13 (USA), was used for the analysis.

## 3. Results

### 3.1. General Results

The study group consisted of twenty-eight PAH patients, with the mean age 49.5 ± 15.5 years (idiopathic/heritable *n* = 19; connective tissue diseases *n* = 4; associated with congenital small/coincidental defects *n* = 5). The control group consisted of twelve healthy volunteers, matched for age and sex (44.75 ± 13.51 years, eight females). Five patients (17%) were incident cases, while the rest of the study group were prevalent and receiving PAH-specific treatment at the time of study. Most of them were in the WHO functional class III (64%, *n* = 18) and the majority of patients were women (60%, *n* = 17). General groups characteristics are presented in Table 1 (the study group was also partially presented in our previous study [10]).

### 3.2. Echo Parameters and RV Function

In the PAH group, RV dimensions were significantly higher than in the control group, (Table 1). The LV ejection fraction did not differ between both groups. The mean estimated mean pulmonary arterial hypertension was 47.29 ± 15.3 mmHg and the mean TAPSE was 19.84 ± 4.28 mm. Only five patients (17%) had severe tricuspid regurgitation. The mean pulmonary vascular resistance in RHC was 9.17 ± 5.69 WU and the cardiac index was 2.65 ± 0.6 L/min/m^2^. There were significant correlations between echo-derived hemodynamic parameters and RHC-derived values, e.g., echo mPAP vs. mPAP (RHC), r = 0.86, *p* < 0.001. In addition, MRI-derived TAPSE values correlated well with echo TAPSE, r = 0.92, *p* < 0.001.

The echo-estimated RV ventricular–arterial coupling surrogate parameter (echo TAPSE/sPAP) was 0.35 ± 0.20 and 1.51 ± 0.22 in the control group, *p* < 0.01. This ratio was comparable to the ratio obtained with a use of MRI (TAPSE) and RHC (sPAP); as shown in the Bland–Altman plot in Figure 1.

We observed that echo TAPSE/sPAP positively correlated with RHC-derived parameters—inversely with the mPAP (r = −0.72, *p* < 0.001), as shown in Figure 2A, PVR (r = −0.61, *p* < 0.001) and positively with cardiac index (r = 0.42, *p* = 0.02). Moreover, the echo TAPSE/sPAP ratio also correlated with other MRI parameters—right ventricle ejection fraction, r = 0.51, *p* = 0.01, as shown in Figure 2B; RV free wall thickness (r = −0.57, *p* = 0.004); and with RV mass index (r = −0.53, *p* = 0.01) in PAH. Echo TAPSE/sPAP significantly correlated with the non-echo ratio of TAPSE (MRI)/sPAP (RHC)—r = 0.75, *p* < 0.005. We did not observe significant correlations of TAPSE/sPAP ratio with other PAH prognostic factors, e.g., 6MWT distance or BNP levels.

### 3.3. Survival Analysis

We analyzed parameters in PAH subjects who had met the combined endpoint—CEP (75%, *n* = 21). After 58 months’ follow-up, six patients died, and fifteen patients had WHO class worsening with hospitalization (including three patients requiring the initiation of a parenteral prostacyclin analogue). The mean time to clinical worsening was 27.16 ± 20.21 months.

Patients who reached CEP had a significantly higher echo mPAP and lower TAPSE (53.11 ± 17.11 mmHg vs. 34.86 ± 8.49 mmHg, *p* = 0.006 and 19.23 ± 4.58 mm vs. 21.17 ± 3.87 mm, *p* = 0.06, respectively), Table 2.

Interestingly, the TAPSE/sPAP ratio was significantly lower in CEP+ patients (0.30 ± 0.21 vs. 0.43 ± 0.23, *p* = 0.04).

ROC analysis confirmed the role of TAPSE/sPAP in predicting CEP (AUC 0.72 (95% CI 0.61–0.92), *p* = 0.02) and revealed the cut-off value −0.25 mm/mmHg. Patients with an echo TAPSE/sPAP lower than 0.25 mm/mmHg had worse prognosis, with log-rank test *p* = 0.001, as shown in Figure 3. All six patients who died also had a TAPSE/sPAP lower than 0.25 mm/mmHg at baseline visit.

Furthermore, ROC analysis revealed that TAPSE and sPAP alone (from echo or MRI) had a lower prediction of CEP than when presented as a ratio (according to the coupling idea), as shown in Table 3. Interestingly, the area under curve for non-echo TAPSE/sPAP (derived from MRI and RHC, respectively) was not significantly different from echo-only TAPSE/sPAP (0.76 [0.59–0.94] vs. 0.72 [0.61–0.92], respectively, *p* = 0.43).

## 4. Discussion

In our study, we confirmed that the simplified, echocardiographic approach of estimating RV–arterial coupling in PAH patients—the TAPSE/sPAP ratio—could be very useful in patient evaluation, even at bedside. It is linked with long-term prognosis and significantly correlates with the hemodynamic parameters of the RV from MRI and RHC.

The idea of coupling is based on the relationship between ventricular contractility and afterload. In case of elevated pulmonary vascular resistance (especially in the early stages of PAH), RV systolic function and afterload are adequately coupled; however, with a further chronic increase in afterload, the RV becomes unable to proportionally enhance its contractility, leading to progressive right heart failure [16,17].

Still, the most accurate protocol for obtaining RV–arterial coupling is direct (invasive) measurement of pressure–volume loops [3,4]. Despite their clinical value, especially in the field of PAH, measurements of effective arterial elastance and end-systolic elastance from pressure–volume loops are not routinely obtained, due to technical difficulties. The possibility of a feasible, echocardiographic approach that mirrors relevant pathophysiological RV mechanisms was recently explored for PAH. For instance, TAPSE/mPAP or fractional area change (FAC)/mPAP were found to be related to established prognostic markers of RV function [18]. According to the coupling idea, the TAPSE/sPAP ratio combines longitudinal shortening with the ability of the RV to generate pulmonary pressure. Recently, the TAPSE/sPAP ratio as a noninvasive surrogate of the RV–arterial coupling in left heart failure was confirmed by Guazzi et al. [5]. This study revealed TAPSE/sPAP as an independent predictor of event-free survival, with a value <0.36 mm/mmHg indicating a particularly poor prognosis, whereas in our study, this value was <0.25 mm/mmHg (in different study population). In another paper, Tello et al. managed to perform invasive pressure–volume loop-derived RV–arterial coupling and compare the results to echo TAPSE/sPAP [18].

In our study, we validated echo TAPSE/sPAP versus MRI- and RHC-derived parameters. As expected, the echo TAPSE/sPAP ratio strongly correlated with the hemodynamic afterload-dependent parameters, e.g., PVR or the cardiac index, which have been separately related to outcome in patients with PAH [1,19]. The current study presents for the first time a direct comparison of ROC curves for predicting CEP for echo TAPSE/sPAP and non-echo TAPSE/sPAP. The results presented above indicate that the TAPSE/sPAP ratio obtained non-invasively provides non-inferior prognostic information even in near-five-year observation.

It is noteworthy that echo, MRI and invasive measurements were performed during the baseline visit (at stable state of PAH patients), and we observed significant correlations between those parameters (Figure 2). This confirms that data used in survival analysis are reliable and represent the actual patients’ state. The presented survival analysis suggests that impaired echo-estimated RV–arterial coupling may precede significant clinical deterioration. This technique could be routinely used during periodic monitoring visits of PAH patients.

It has to be mentioned that our study group consisted mostly of patients with idiopathic PAH in WHO III class with severely deteriorated pulmonary hemodynamics (the mean mPAP of the study group was >40 mmHg, with the border cardiac index). The TAPSE/sPAP ratio was low overall, and therefore, we cannot directly transfer our results to patients with less severe PAH. On the other hand, patients with the lowest TAPSE/sPAP values at baseline had a worse prognosis. This could help in risk stratification by detecting the group of PAH patients who need therapy escalation in a shorter period of time. We are aware that sPAP is an imperfect measure of RV afterload. By definition, it combines components of both vascular resistance and pulmonary venous congestion [16], and a better option would be the pulmonary artery compliance parameter, but this requires invasive measurements. Our main goal was to validate the proposed noninvasive surrogate of RV–arterial coupling, as it seems that the confirmation of subtle changes in RV coupling may be crucial for PAH patients’ prognosis.

There are also limitations on a use of TAPSE as a measure of RV systolic function. TAPSE provides information on the longitudinal motion of the chamber (which reflects contractility), but it is highly angle- and operator-dependent [20]. In the case of severe tricuspid regurgitation, its reliability is limited; thus, TAPSE values may be overestimated. In our study, only five patients had severe tricuspid regurgitation, but it has to be mentioned that the usefulness of TAPSE may be significantly reduced in end-stage RV failure. In addition, the study group included patients with congenital heart diseases. This group of PAH patients has a better overall prognosis and survival rate, which may have had an impact on the results. Therefore, we suggest using the TAPSE/sPAP ratio merely as an additional tool during complex baseline PAH patient assessment, especially in already high-risk groups of PAH patients (with initially high pulmonary pressures). This additional parameter may help to detect subgroups of patients in need of rapid specific therapy escalation or qualification for lung transplantation.

TAPSE is also an afterload-dependent parameter and itself is an important predictor of poor outcome in PAH [21,22]. On the other hand, the TAPSE/sPAP ratio may be also defined as an attempt to normalize TAPSE for sPAP as an index of load adaptability; thus, this indexation theoretically creates a better parameter, specially based on the idea of coupling. Finally, when we compared ROC curves for predicting CEP, the AUC for the TAPSE/sPAP ratio is significantly higher than for TAPSE or sPAP alone.

## 5. Conclusions

The proposed echocardiographic estimation of RV ventricular–arterial coupling offers a potential easy, reliable and non-invasive prognostic parameter for the more comprehensive assessment of hemodynamic adaptation in patients with pulmonary arterial hypertension, even at bedside. It is significantly related with other invasive measurements or MRI-derived parameters. Identifying PAH patients with a nearly uncoupled right ventricle before clinical deterioration may be crucial to determining their prognosis.

## Figures and Tables

**Figure 1 jcm-10-02995-f001:**
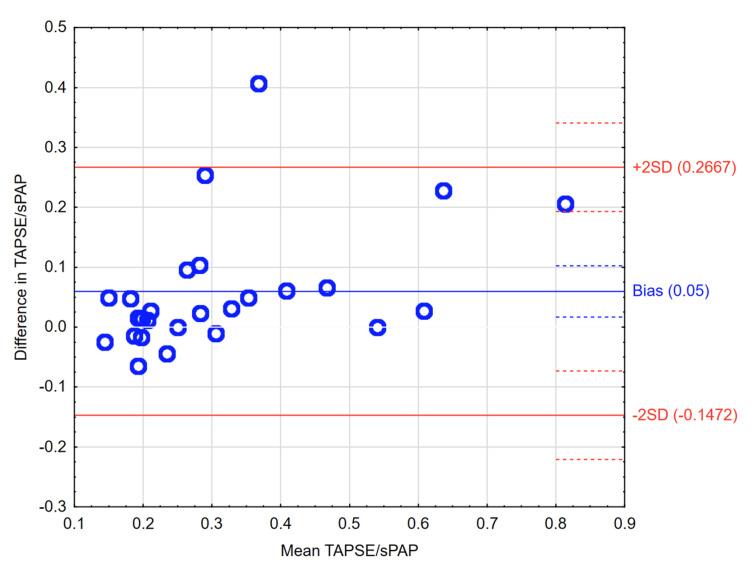
Bland–Altman plot presenting differences between echo-derived TAPSE/sPAP and non-echo-derived TAPSE (MRI)/sPAP (RHC) ratios.

**Figure 2 jcm-10-02995-f002:**
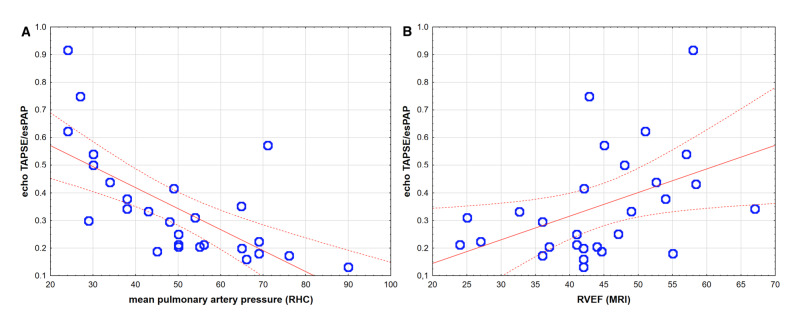
Spearman correlations between echo TAPSE/sPAP ratio and (**A**) mean pulmonary artery pressure, mPAP (r = −0.72, *p* < 0.001) and (**B**) right ventricle ejection fraction, RVEF (r = 0.51, *p* = 0.01).

**Figure 3 jcm-10-02995-f003:**
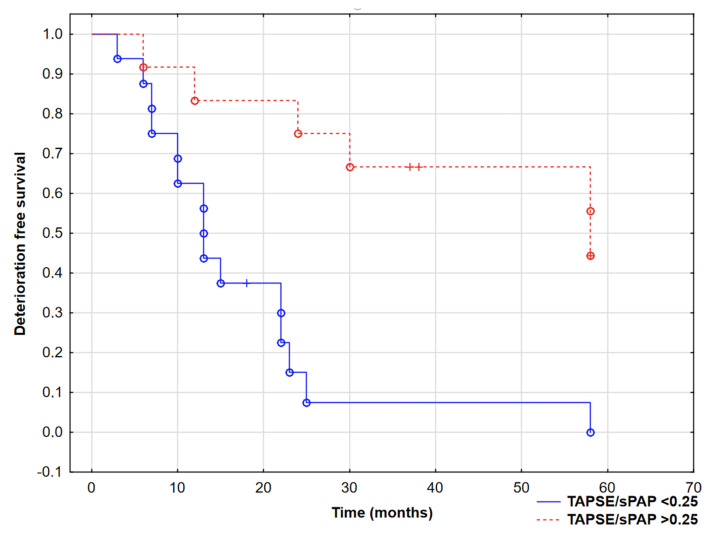
Kaplan–Meier curves presenting deterioration-free survival in PAH patients based on TAPSE/sPAP ratio, log-rank test, *p* = 0.0004. °—complete events, +—censored events.

**Table 1 jcm-10-02995-t001:** General characteristics of pulmonary arterial hypertension (PAH) patients and control group.

	PAH Group	Control Group
Patients, *n*	28	12
Age (years)	49.5 ± 15.5	44.75 ± 13.51
Female sex, % (*n*)	60 (17)	67 (8)
BMI (kg/m^2^)	24.5 ± 3.7	25.2 ± 3.6
6MWD (m)	392 (322–453) *	525 (497–550)
BNP (pg/mL)	310.8 (54–3654) *	27.2 (13–42)
Aetiology		
IPAH/HPAH, % (*n*)	67 (19)	
CTDPH, % (*n*)	14 (4)	
CHDPH, % (*n*)	19 (5)	
Therapy		
PDE5 inhibitors, % (*n*)	38 (10)	
Prostacyclins, % (*n*)	20 (5)	
ERA, % (*n*)	11 (3)	
Dual PDE5 inhibitor + ERA, % (*n*)	31 (8)	
Echo parameters		
RV basal diameter (cm)	6.6 *	3.6 ± 0.6
RAA (cm^2^)	25 (19–32) *	10 (9–12)
TAPSE (mm)	19.84 ± 4.28 *	26 ± 3.4
emPAP (mmHg)	47.29 ± 15.3 *	12.2 ± 3.8
AcT (ms)	82 ± 22 *	130 ± 28
TAPSE/sPAP (echo)	0.34 ± 0.19 *	1.51 ± 0.22
RHC parameters		
sPAP, mmHg	79.6 ± 30.72	
dPAP, mmHg	31.9 ± 13.54	
mPAP, mmHg	48.95 ± 18.77	
PAWP, mmHg	10.5 ± 2.03	
DPG, mmHg	21.85 ± 13.08	
CI, L/min/m^2^	2.65 ± 0.60	
RAP, mmHg	8.2 ± 2.92	
PVR, Wood Units	9.17 ± 5.69	
MRI parameters		
RVEF, %	44.91 ± 7.9 *	63.8 ± 5.8
RV EDV/BSA, mL/m^2^	118.2 ± 21.7 *	73.6 ± 12.2
RV ESV/BSA, mL/m^2^	65.9 ± 20.3 *	28.2 ± 9.6
RV mass/BSA, g/m^2^	38.8 ± 13.9 *	23.8 ± 4.9
LVEF, %	60.3 ± 9.9	67.1 ± 4.4
LV mass/BSA, g/m^2^	58.3 ± 14.4	59.7 ± 18.3
RV thickness, mm	5.8 ± 1.5 *	2.58 ± 0.4
IVS, mm	8.3 ± 1.5	7.7 ± 1.2
TAPSE, mm	18.9 ± 4.4 *	24.9 ± 2.9

Data are presented as mean ± standard deviation or median (interquartile range). *—statistically significant difference between PAH and control groups, *p* < 0.05. 6MWD, 6 min walk test distance; AcT, acceleration time; BMI, body mass index; BSA, body surface area; BNP, brain natriuretic peptide; CI, cardiac index; CHDPAH, congenital heart disease related pulmonary arterial hypertension; CTDPAH, connective tissue disease related pulmonary arterial hypertension; DPG, diastolic pulmonary gradient; dPAP, diastolic pulmonary artery pressure; EDV, end-diastolic volume; emPAP, echo mean pulmonary artery pressure; ERA, endothelin receptor antagonist; ESV, end-systolic volume; HPAH, heritable pulmonary arterial hypertension; IPAH, idiopathic pulmonary arterial hypertension; IVS, interventricular septum; LV, left ventricle; LVEF, left ventricle ejection fraction; mPAP, mean pulmonary artery pressure; MRI, magnetic resonance imaging; PAH, pulmonary arterial hypertension; PASP, pulmonary artery systolic pressure; PAWP, pulmonary artery wedge pressure; PDE5, phosphodiesterase type 5; PVR, pulmonary vascular resistance; sPAP, systolic pulmonary artery pressure; RAA, right atrial area; RAP, right atrial pressure; RHC, right heart catheterization; RV, right ventricle; RVEF, right ventricle ejection fraction; SUV, standardized uptake value; SvO2, mixed venous oxygen saturation; TAPSE, tricuspid annular plane systolic excursion; WHO, World Health Organization.

**Table 2 jcm-10-02995-t002:** Comparison of PAH patients with combined endpoint (CEP) and without CEP.

	CEP (+) Patients	CEP (−) Patients	*p*-Value
Patients, *n*	21	7	
BNP, pg/mL	269 (73–410)	62 (46–258)	0.60
6MWT distance, m	394.9 *±* 92.5	413.6 *±* 71.4	0.04
Echo parameters			
LVIDd, cm	4.38 *±* 0.39	4.57 *±* 0.55	0.50
IVSd, cm	0.93 *±* 0.08	1.24 *±* 0.82	0.76
PWd, cm	0.91 *±* 0.13	0.90 *±* 0.17	0.79
RVIDd, cm	3.44 *±* 0.59	3.41 *±* 0.51	0.81
RAA, cm^2^	24.72 *±* 8.63	20.42 *±* 6.8	0.27
sPAP, mmHg	72.86 *±* 32.74	58.50 *±* 21.78	0.12
mPAP, mmHg	53.11 *±* 17.11	34.86 *±* 8.49	0.03
TAPSE, mm	19.23 ± 4.58	21.17 ± 3.87	0.06
AcT, ms	98.81 *±* 32.2	100.71 *±* 26.49	0.72
TAPSE/sPAP	0.30 ± 0.17	0.43 ± 0.21	0.04
RHC parameters			
sPAP, mmHg	85.44 *±* 28.11	62.61 *±* 18.63	0.04
mPAP, mmHg	54.94 *±* 16.95	37.21 *±* 10.84	0.006
PVR, Wood Units	10.39 ± 5.59	5.74 ± 3.3	0.01
CI, L/min/m^2^	2.49 ± 0.66	2.92 ± 0.84	0.01
MRI parameters			
RVEF, %	42.05 ± 9.53	50.32 ± 9.11	0.01
RV mass/BSA, g/m^2^	46.27 *±* 17.41	36.71 ± 14.25	0.04
RV thickness, mm	6.22 *±* 1.35	4.8 ± 1.2	0.04
IVS, mm	8.55 ± 1.54	8.77 ± 2.43	0.71
TAPSE, mm	17.72 ± 4.14	21.75 ± 3.73	0.04

Data are presented as mean ± standard deviation or median (interquartile range). 6MWD, 6 min walk test distance; AcT, acceleration time; BMI, body mass index; BSA, body surface area; BNP, brain natriuretic peptide; CI, cardiac index; DPG, diastolic pulmonary gradient; dPAP, diastolic pulmonary artery pressure; EDV, end-diastolic volume; emPAP, echo mean pulmonary artery pressure; ESV, end-systolic volume; IVS, interventricular septum; LV, left ventricle; LVEF, left ventricle ejection fraction; mPAP, mean pulmonary artery pressure; MRI, magnetic resonance imaging; PAH, pulmonary arterial hypertension; PASP, pulmonary artery systolic pressure; PAWP, pulmonary artery wedge pressure; PVR, pulmonary vascular resistance; sPAP, systolic pulmonary artery pressure; RAA, right atrial area; RAP, right atrial pressure; RHC, right heart catheterization; RV, right ventricle; RVEF, right ventricle ejection fraction; SUV, standardized uptake value; SvO2, mixed venous oxygen saturation; TAPSE, tricuspid annular plane systolic excursion; WHO, World Health Organization.

**Table 3 jcm-10-02995-t003:** Comparison of area under curve (AUC) of various parameters for prediction of combined endpoint (ROC analysis).

Parameter	AUC (95% Confidence Interval)	*p*-Value
sPAP (echo)	0.71 (0.48–0.94)	*p* = 0.06
sPAP (RHC)	0.70 (0.55–0.92)	*p* = 0.01
TAPSE (echo)	0.65 (0.39–0.92)	*p* = 0.21
TAPSE (MRI)	0.67 (0.38–0.95)	*p* = 0.23
TAPSE/sPAP (echo)	0.72 (0.61–0.92)	*p* = 0.02
TAPSE/sPAP (MRI/RHC)	0.76 (0.59–0.94)	*p* = 0.003
PVR (RHC)	0.71 (0.50–0.91)	*p* = 0.04

MRI, magnetic resonance imaging; PVR, pulmonary vascular resistance; RHC, right heart catheterization; sPAP, systemic pulmonary artery pressure; TAPSE, tricuspid annular plane systolic excursion.

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
