# Peer review of "Echocardiographic Assessment of Right Ventricular–Arterial Coupling in Predicting Prognosis of Pulmonary Arterial Hypertension Patients"

_jcm, 2021, doi:10.3390/jcm10132995_

Round 1

Reviewer 1 Report

TAPSE/PASP ratio is known to be a meaningful prognostic parameter in patients with PAH and is associated with hemodynamics and functional class. in this article, it seems to be a prognostic index associated with hemodynamic and MRI-derived parameters. 

The article is well written however the cohort is small (28 patients). Moreover, 19% of them are Congenital Heart Disease-PAH patients, which probably offers a bias in the results as this population is proved to have better overall prognosis and survival. this could be mention as as a limitation of the study.

in line 254, the authors mention the cohort is predominantly IPAH with severely compromised hemodynamic parameters. however, in table 1, the median CI is reported to be 2.65 l/min/m2 representing an increased value and low-risk patients according to ESC/ERS guidelines for Diagnosis and Treatment of PH. CPE+ patients had also an increased mean CI: 2.49 l/min/m2. so, probably it could be better to delete the two sentences in lines254-257. This could be also consistent with the conclusion as authors say 'selecting PAH patients with nearly uncoupled right ventricle before clinical deterioration may be crucial to determine their prognosis'. 

Author Response

We would like to thank the Reviewer for important comments. We agree that including into the study group Congenital Heart Disease-PAH patients may “slightly improve” prognosis. However, it should be stressed, that we excluded Eisenmenger physiology as well as PAH associated with prevalent systemic-to-pulmonary shunts due to moderate to large defects (according to European guidelines), hence the major pathophysiology in the study group was consistent. Nevertheless, we added following sentence to the manuscript (in study limitation section):

Also study group included patients with congenital heart diseases. This particular group of PAH patients has better overall prognosis and survival, what may have an impact on the result. Therefore, we suggest using TAPSE/sPAP ratio rather as an additional tool during complex baseline PAH patient’s assessment, especially in already high-risk groups of PAH patients (with initially high pulmonary pressures).”

Considering Cardiac Index values – we agree that mean CI places an “average patient” in a low-risk group according to ESC Guidelines. However, an average of 2.49 l/min/m2 in CPE+ patient is below the normal values. One has to bear in mind a high proportion of patients with congenital heart disease in whom there is increased CI as well as 20% of patients on prostanoids, who also often present increased CI. Due to limitations of our Ethical Committee consent we did not perform RHC in the control group to present this in the paper.

To be clear - we used the terms “before clinical deterioration” and “group of already severe PAH” to underline that patients during the baseline visit were mostly in WHO III Class, with severely elevated pulmonary pressures (mPAP>40mmHg) (Table 1) but with still preserved CO/CI thus they were not hospitalized at that moment. In our opinion, it is indeed the situation when physicians may use additional, newly proposed parameters to determine whether patients even with border CI remain “stable” (not hospitalized/no PAH therapy escalation) or not.

We added following sentence to the manuscript in line 261-262:

“It has to be mentioned that our study group consisted mostly of patients with idiopathic PAH in WHO III class with severely deteriorated pulmonary hemodynamics (mean mPAP of study group was > 40mmHg, with border cardiac index). The TAPSE/sPAP ratio was low overall and therefore we cannot directly transfer our results to patients with less severe PAH. On the other side, patients with the lowest TAPSE/sPAP values at baseline had worse prognosis.”

Yours sincerely,

Karol Kamiński

Reviewer 2 Report

The paper entitled “Echocardiographic assessment of right ventricular-arterial coupling in predicting prognosis of pulmonary arterial hypertension patients.” by Dr. Remigiusz Kazimierczyk, et al. investigates the usefulness of TAPSE/sPAP evaluated by echocardiography in PAH. The authors investigated the relationships between TAPSE/sPAP and the combined endpoint. I consider that this article is well-written and descriptive. This topic is relevant to the field of this journal, and the article can be quite informative and educational to the readers.  

In this study, the echocardiographic assessment of RV-arterial coupling in PAH could be very useful in patient’s evaluation. TAPSE/sPAP is linked with long-term prognosis and significantly correlates with hemodynamic parameters. Although there are several previous reports regarding the clinical utility of TAPSE/sPAP, this manuscript revealed the usefulness of this index in this different population. 

Author Response

We would like to thank the Reviewer for the comment.

This manuscript is a resubmission of an earlier submission. The following is a list of the peer review reports and author responses from that submission.